# Isolation of a Novel Caprine *Eimeria christenseni* Strain (GC) in Canary Islands and Analysis of Parasitological, Clinical, and Pathological Findings on Experimentally Infected Goat Kids

**DOI:** 10.3390/ani15020139

**Published:** 2025-01-08

**Authors:** Emilio Barba, José Manuel Molina, Francisco Rodríguez, Otilia Ferrer, María Carmen Muñoz, Liliana M. R. Silva, María Cristina Del Río, José Adrián Molina, Anja Taubert, Carlos Hermosilla, Antonio Ruiz

**Affiliations:** 1Department of Animal Pathology, Faculty of Veterinary Medicine, University of Las Palmas of Gran Canaria, 35001 Las Palmas de Gran Canaria, Spain; emilio.barba101@alu.ulpgc.es (E.B.); josemanuel.molina@ulpgc.es (J.M.M.); otila.ferrer@ulpgc.es (O.F.); mariadelcarmen.munoz@ulgpc.es (M.C.M.); jose.molina108@alu.ulpgc.es (J.A.M.); 2Department of Anatomy and Compared Anatomy Pathology, Faculty of Veterinary Medicine, University of Las Palmas of Gran Canaria, 35001 Las Palmas de Gran Canaria, Spain; francisco.guisado@ulpgc.es; 3Faculty of Veterinary Medicine, Institute of Parasitology, Justus Liebig University Giessen, 35390 Giessen, Germany; liliana.silva@vetmed.uni-giessen.de (L.M.R.S.); anja.taubert@vetmed.uni-giessen.de (A.T.); carlos.r.hermosilla@vetmed.uni-giessen.de (C.H.); 4Egas Moniz Center for Interdisciplinary Research (CiiEM), Egas Moniz School of Health & Science, 2829-511 Caparica, Portugal

**Keywords:** coccidia, goats, new strain, *Eimeria christenseni*, experimental infection

## Abstract

Coccidiosis is one of the most economically important parasitic diseases in goat production systems. The disease is caused by protozoa, referred to as coccidia, of which *Eimeria christenseni* is among the most pathogenic. The aim of this study was to isolate an *E. christenseni* strain and to assess its infectivity, pathogenicity, and ability to develop a protective immune response. After previous collection of *E. christenseni*-positive faeces, purification of oocysts, and amplification in donor animals, an experimental infection was carried out considering three groups: primary-infected, reinfected, and non-infected. The results showed that the Gran Canaria (GC) *E. christenseni* strain had similar morphological and biological characteristics to those previously described, but no significant clinical signs were observed despite the high number of dissemination parasitic elements (oocysts) excreted with faeces. The novel strain isolated would therefore be of low pathogenicity but still able to protect hosts from subsequent reinfections. Its biological similarities to other highly pathogenic *Eimeria* species might enable comparative studies aimed at developing alternative strategies to control goat coccidiosis.

## 1. Introduction

Coccidiosis is one of the most important parasitic diseases affecting goats, and billions of USD/EUR are annually invested worldwide to minimise its effects [1]. The species causing coccidiosis in goats are host-specific (i.e., monoxenous) [2], with *Eimeria ninakohlyakimovae* and *Eimeria arloingi* being the most pathogenic [3], while *Eimeria caprina*, *Eimeria christenseni*, and *Eimeria alijevi* are considered of moderate pathogenicity [4,5]. Regardless of age, animals may be infected by more than one different *Eimeria* species at the same time [6,7], and mixed infections are the rule worldwide [2]. However, cell-mediated protective immunity against *Eimeria* spp. is reported as species-specific [8,9] or even strain-specific [10,11,12,13]. Therefore, protective cross-immunity is rare, and immunological studies on specific *Eimeria* species in the respective host are necessary.

The life cycle of *Eimeria* spp. consists of an exogenous phase (sporogony) and an endogenous phase, which comprises asexual (merogony or schizogony) and sexual (gamogony) stages [3,14]. Like the bovine system, where the most pathogenic species develop macromeronts in the endothelial cells of lacteal ducts in the ileum, pathogenic goat species also develop macromeronts in this particular location [15,16], which in some cases leads to severe intestinal disease characterised by catarrhal diarrhoea, weight loss, dehydration, and growth delay [17]. As stated above, *E. christenseni*, an *Eimeria* species recently described as highly prevalent in goat-rearing systems [18], also develops macromeronts in endothelial cells of the same intestinal section [19]. The first generation of schizonts is immunologically important for several reasons. Firstly, all these pathogenic ruminant *Eimeria* species can immunomodulate their host endothelial cells by modifying the cell cycle [20,21], the cytoskeleton [22], cellular metabolism and senescence [21], nutrients [20,22,23], and even apoptosis [24,25]. Secondly, there is some evidence in the literature that *Eimeria* first-generation meronts represent a major target for protective immune reactions [26]. In agreement, it has been reported that protective immune responses during prepatency in goat kids experimentally infected with *E. ninakohlyakimovae* can be attributed to first-generation macroschizonts in vivo [27]. In general, apart from innate immune responses, the termination of *Eimeria* spp. primary infections, as well as the control of homologous challenge infections in ruminants, is based on host cellular adaptive immune reactions [27,28], involving different immunocompetent cells [27,29,30] and cytokines [27,31].

The control of coccidiosis has traditionally relied on prophylactic or metaphylactic treatments with anticoccidials, but important efforts are being made to develop alternative strategies, including phytotherapy [32,33,34] and immunoprophylaxis [29,35]. All these approaches require standardised methods for efficacy testing as suggested by the WAAVP guidelines [36]. The authors give special emphasis to the need for strictly defined laboratory strains of mammalian coccidia derived from single-oocyst infections that are maintained routinely in experimental research units, as reported for *E. bovis* (strain H; [9]), *E. arloingi* (strain A [16]), and *E. ninakohlyakimovae* (strain GC; [37]). In general, due to the current lack of research on cellular immune responses against *Eimeria* species affecting goats, there is a need to isolate *Eimeria* spp. strains for basic research and for the possible development of strategies for prophylaxis and control of this very common disease in goats.

As referred to above, *E. christenseni* has important similarities to *E. ninakohlyakimovae* and *E. arloingi* in relation to life cycle at the first schizogony within endothelial cells and comparative studies between those species would be highly interesting for both basic and immunological research. Specifically, the mechanisms and pathogenesis and associated immune response would be of great scientific interest considering the lower pathogenicity of *E. christenseni* compared to the other species. To address this issue, the present study was conducted to not only isolate a novel field strain of *E. christenseni* but also to evaluate its infectivity, pathogenicity, and development of protective immunity in experimentally infected goat kids.

## 2. Materials and Methods

### 2.1. Ethical Statement

All animal procedures were carried out in strict accordance with national ethics, current European legislation on animal welfare (ART13TFEU), and protocols approved by the institutional review board (OEBA-ULPGC 15/2019R1).

### 2.2. Animals

Goat kids of the Majorera breed, purchased from a local farmer at the age of 1 to 3 days old, were used. This breed, originally from Fuerteventura (Canary Islands, Spain), is characterised by short hair and large head with arched horns and is mainly a dairy breed. It represents the goat breed with the largest population in the Canary Islands [38].

For the experimental infections, the animals were individually identified by means of ear tags and moved to the Experimental Animal Facility of the Faculty of Veterinary Medicine (Scientific Park, University of Las Palmas de Gran Canaria). Upon arrival, the goat kids were washed and dried to eliminate any oocysts that might be adhered to them and thereafter placed in clean and disinfected metabolic boxes. On day 1 of arrival, a single dose of 1 mg/kg diclazuril (Rumicox, Esteve, Barcelona, Spain) was given to the animals, and additionally, they were treated with Halocur (MSD Animal Health, Salamanca, Spain) for one week. During the first 2 weeks of life, the kids were fed with a milk replacer (Bacilactol, Capisa, Canarias, Spain), with the gradual addition of commercial pellets (Starting Concentrate, Capisa, Canarias, Spain). Water and sterilised hay were given ad libitum. During the experiments, all necessary precautions were taken to prevent unwanted infections, particularly caprine Eimeria infections.

A total of 22 animals (*n* = 22) were used, 3 for isolation of the *E. christenseni* field strain and 19 additional goat kids for experimental studies on infectivity, pathogenicity, and immune response of the new strain.

### 2.3. Parasites

To obtain the novel *E. christenseni* strain (GC strain), faecal samples were collected by rectal route from naturally infected goat kids, and afterwards, donor animals were infected for oocyst amplification following the protocol described by Silva and Lima [39]. Faeces containing at least 90% purity for *E. christenseni* oocysts were floated for 40 min in a saturated sodium chloride solution of 1.19 g/L density. The collected oocysts were incubated in a 2% potassium dichromate solution (K_2_Cr_2_0_7_) in a Petri dish for a week at room temperature (25 °C), shaking the suspension daily to infuse oxygen to facilitate the sporulation. To determine the sporulation time under these experimental conditions, the incubation sample was monitored daily, and the percentage of sporulation was calculated. Sporulated oocysts were collected and maintained at 4 °C in culture flasks containing 2% potassium dichromate. Initially, a total of 1.3 × 10^6^ oocysts was obtained, with a purity for *E. christenseni* of 90%. The remaining 10% included other species, notably *E. ninakohlyakimovae* and *E. arloingi*. The identification of *Eimeria* species was performed under a 40× objective using a calibrated eyepiece according to previously reported keys [40,41,42].

For amplification and to improve the purity percentage, the initial batch of oocysts was used to infect 3 goat kids aged 3 weeks. To optimise the infecting dose, each animal was infected with a different number of oocysts: 5 × 10^4^ oocysts (animal 1), 1 × 10^5^ oocysts (animal 2), and 2 × 10^5^ oocysts (animal 3). Prior to infection, oocysts were washed 3 times with distilled water to remove potassium dichromate, and the infection was performed orally using a 5 mL syringe.

Faecal samples were collected individually starting on day 12 post infection (p. i.) and for 16 consecutive days. Oocyst counts per gram of faeces (OPG) were determined using the modified McMaster technique [43], with a sensitivity of ≥100 OPG. Faecal samples with higher OPG counts were pooled daily, and the corresponding oocysts were isolated according to Jackson [44] with some modifications. The faeces were mixed 1:1 with water and filtered through sieves of decreasing pore diameter to approximately 100 µm. The faecal mixture was subsequently mixed 1:1 with saturated sugar solution and left to float on a glass slide, which was washed with distilled water every 2 h for three consecutive days. The collected washes were centrifuged at 2300× *g* for 20 min, the supernatants were discarded, and the resulting sediments were mixed and transferred to a glass Erlenmeyer flask. As described above, the oocysts were then incubated in a 2% potassium dichromate that was adjusted to sporulate under constant aeration for 7 days at room temperature (RT). Sporulated oocysts were stored at 4 °C in culture flasks (Nunc) with 2% potassium dichromate and air access. The percentage of purity for *E. christenseni* and the sporulation rate were determined for the different faecal batches processed.

### 2.4. Morphological Characterisation of the Novel Eimeria christenseni Strain

The morphological characteristics of *E. christenseni* oocysts were established based on the same oocyst batch employed in the first donor animal infection. By analysing 200 oocysts and 200 sporozoites, the following parameters were determined: oocyst longitudinal diameter, oocyst transversal diameter, micropillar cap thickness, oocyst membrane colour, sporozoite longitudinal diameter, and sporozoite transversal diameter. For this purpose, direct observations of oocyst samples were performed with an Olympus microscope (Olympus CH40, Barcelona, Spain) and associated digital camera (motican BTW, Hong Kong, China) by using 100× and 400× magnifications. The program Moti Connect^®^ was employed for measurement and picture recording.

### 2.5. Experimental Design

A total of 19 goat kids (*n* = 19) were divided into 3 groups: G1, animals infected with sporulated *E. christenseni* oocysts at two weeks of age and challenge-infected at six weeks of age (*n* = 7); G2, animals primary-infected at six weeks of age or challenge control (*n* = 6); G3, non-infected control animals (n = 5). The oocysts selected for infection belonged to samples from the pool obtained from the three donor animals on days 26–28 p. i., and their purity was 99%. Based on the outcome of infection in donor animals, an infecting dose of 2 × 10^5^ sporulated oocysts of *E. christenseni* GC strain was used in both primary and challenge infections. All the animals were infected orally by using a syringe. The different groups were placed in separate metabolic cages and kept in parasite-free conditions, being handled with a strict hygienic protocol. As described in Section 2.6, the level of immunoprotection of the *E. christenseni* GC strain, its infectivity, and its pathogenic effects were evaluated by productive (body weight), clinical (presence of clinical signs and variations in blood parameters), parasitic (oocyst counts in faeces), and pathological (macro- and microscopic intestinal lesions) analyses.

### 2.6. Coprological, Clinical, and Histopathological Analyses

For coprological analysis, faecal samples were obtained daily from 14 days postinfection (d. p. i.; coinciding with the first oocyst detection recorded in donor goat kids) until the animals were slaughtered. The control group also underwent coprological analysis once a week to verify the absence of infection. Faecal samples were obtained from the rectum, weighed, and quantified by a modified McMaster method [43] to determine oocysts per gram of faeces (OPG) counts. As a variation of this technique, in cases where the amount of stool was less than a gram, a different volume of saturated NaCl solution was employed: 30 mL was used for samples weighing more than 0.4 g, 10 mL between 0.2 and 0.4 g, and 5 mL below 0.2 g. Afterwards, the corresponding calculations were made to express the result in OPG counts. If necessary, in the presence of a very high number of oocysts in the faecal samples, a dilution of the samples was carried out (from 10 to 1000 times) to perform an adequate count [45]. In addition, once the count was performed, a flotation in a saturated NaCl solution and subsequent analysis were conducted to verify that the oocysts corresponded to *E. christenseni*.

As for clinical parameters, the presence of diarrhoea and variations in blood parameters were evaluated. The characterisation of the type of diarrhoea was determined by coinciding with the faecal sampling following the score previously described [29,37]: (1) normal consistency; (2) soft unformed; (3) semiliquid; (4) watery diarrhoea; (5) diarrhoea with blood and/or fragment of intestinal mucosa. The body weight of the animals of all groups was weekly determined from week of challenge infections (day 28 of the experiment) until the end of this study. At the same intervals, haematological analyses were performed on blood samples taken by puncture of the vena jugularis. Total and differential leukocyte counts and haematocrit concentrations were determined by using a ProCyte Dx^®^ haematology analyser (IDEXX, Barcelona, Spain). Values of serum total proteins were calculated according to standard procedures by centrifugation of capillary tubes in a microhaematocrit centrifuge.

Pathological analyses were carried out after the animals were euthanised and subsequently submitted to necropsy. All macroscopic lesions were analysed, and different samples were collected from the intestinal mucosa (i.e., ileum, jejunum, and caecum) and mesenteric lymph nodes. Tissue samples were fixed in 10% formalin and embedded in paraffin. Transverse sections of 4–5 µm were stained with haematoxylin and eosin (HE) according to standard staining procedures and then analysed microscopically. Afterwards, the following parameters were evaluated: (1) presence of microabscesses; (2) intensity of infection; (3) presence of necrosis; (4) follicular lymphoid hyperplasia of the ileum; (5) hyperaemia; (6) mucosal thickness; (7) inflammatory cell counts; (8) follicular lymphoid hyperplasia of the mesenteric lymph nodes. Microabscesses, infection intensity, necrosis, and hyperaemia were qualitatively scored as described in paragraph 3.2.3. Ileal lymphoid follicular size, mucosal thickness, and lymphoid follicular size of mesenteric lymph nodes were measured by using a millimetre eyepiece with a 40× objective. Finally, inflammatory cell counts were performed over a total of 8 fields by using a 40× objective and expressed as cells/mm^2^. Parameters measured and/or quantified were further scored as described below. Ileal follicular size: (0) <3 µm; (1) 3–4 µm; (2) 4–5 µm; (3) >5 µm. Mucosal thickness: (0) <500 µm; (1) 500–600 µm; (2) 600–700 µm; (3) >600 µm. Follicular size of mesenteric lymph nodes: (0) <3 µm; (1) 3–4 µm; (2) 4–5 µm; (3) >5 µm. Inflammatory cell count: (0) <50 cells; (1) 50–100 cells; (2) 100–150 cells; (3) >150 cells.

### 2.7. Statistical Analysis

Faecal oocyst counts were logarithmically transformed (log [OPG + 1]), and a Kolmogorov–Smirnov normality test was employed for checking if variables were normally distributed. For the estimation of body weight improvements, the data were additionally analysed as growth rate (ln weight–2 ln weight 1)/t × 100), with t representing the number of days between the sampling time points 1 and 2. One factorial analysis of variance and Tukey’s multiple comparison test were employed to check statistical differences on OPG counts, body weights, and histopathological parameters between groups, while the non-parametric Chi-square was used to analyse the sum of the different scores established for histopathological parameters. The software SigmaPlot 14.5 (Los Angeles, CA, USA) was employed for all the statistical analyses.

## 3. Results

### 3.1. Morphological Characteristics of Eimeria christenseni (GC Strain) Oocysts

Daily monitoring of the exogenous oocyst sporulation rate (i.e., sporogony) revealed that the average time for full sporulation of *E. christenseni* oocysts was 7 ± 1 days. Only full sporulated oocysts were employed for the morphological characterisation.

The oocysts of *E. christenseni* had a medium size of 40.5 × 27.3 µm (Table 1), with a mean length–width radio of 1.48, and showed an oval (sometimes “pear-shape”) morphology (Figure 1A). Sporocysts had an elongated “pear” shape and a mean size of 16.0 × 9.6 µm. Some oocysts had a yellow-brown tone in their wall, while most of them seemed to be yellow-green in colour. The inner layer of the wall had a bright orange tone, sometimes also present in the thickness of the whole oocyst wall. The micropillar cap had an average thickness of 2.1 µm and sometimes was partially detached (Figure 1B). Detachment could be total, so some oocysts without micropillar cap could be observed during microscopic examinations (Figure 1C).

### 3.2. Characterisation of Eimeria christenseni (GC Strain) in Primary and Challenge Infections

#### 3.2.1. Oocyst Excretion Pattern (Figure 2)

Primary-infected goat kids at 2 weeks of age (G1: group 1) began to excrete oocysts on day 15 p. i., with the prepatent period in this group ranging from 15 to 19 days p. i. depending on the animal. Mean OPG levels remained relatively stable during the first week of oocyst excretion, always with values below 600,000 OPG. From day 22 p. i. onwards, there was a progressive increase in the oocyst release, reaching peak values on day 24 p. i. (5.8 × 10^6^ OPG). Subsequently, oocyst counts progressively decreased, and the lowest OPG count (i.e., 137,227 OPG) was found on day 51 p. i. Following the challenge, which was performed 28 days after the primary infection (see arrow, Figure 2), the levels of oocysts excreted with faeces had a postchallenge peak value on day 52 of the experiment (=25 days p. i.). This peak value (855,556 OPG) was much less pronounced than the one reached after the primary infection (*p* < 0.05). Subsequent oocyst counts decreased progressively until the end of the experiment, with values below 200,000 oocysts being observed on the last day of sampling (61 days of the experiment, 34 days after challenge infection).

**Figure 2 animals-15-00139-f002:**
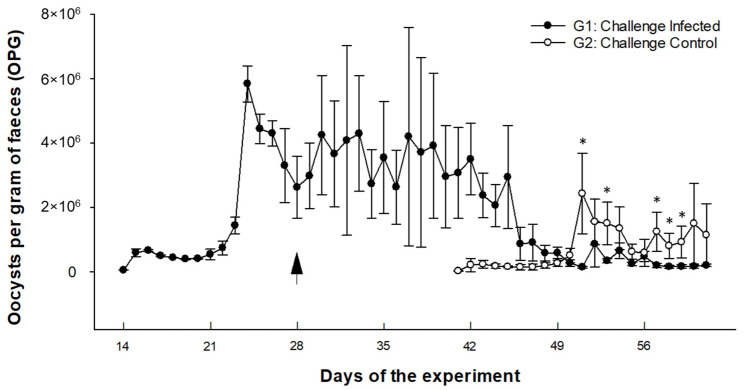
OPG counts in goat kids challenge-infected (G1) and primary-infected (G2) with *E. christenseni* GC strain. The arrow shows the time of challenge infection (G1) or primary infection (G2). Data are expressed as mean ± SEM. (*) *p* < 0.05.

In group 2 (G2), goat kids infected for the first time at 6 weeks of age, the prepatent period ranged from 15 to 17 days p. i. depending on the animal. Oocyst shedding on day 15 p. i. had a mean value of 1.5 × 10^4^ OPG, increased thereafter to 2 × 10^5^ OPG, approximately, and remained at approximately these levels until the day 22 p. i. Subsequently, a further increase was observed, with maximum peak OPG values being recorded on day 25 p. i. (1.5 × 10^6^ OPG). From this point onwards, there was a progressive reduction in the oocyst excretion, with the lowest OPG counts being recorded at the last sampling of the experiment (34 days p. i.).

In the daily examination of individual faecal samples, no oocysts of species other than *E. christenseni* were recorded. On the other hand, faeces from animals in the control group were analysed weekly, and no oocysts were found at any time during the experiment.

#### 3.2.2. Clinical and Productive Parameters

Faecal consistency was evaluated daily with a score previously set between 1 and 5. Throughout this study, all the animals showed score values very close to 1 (completely solid-formed faeces), and no significant differences were observed among the three experimental groups.

With respect to haematological parameters, all goat kids remained within the reference values established for goats, although small differences were observed between the experimental groups and the control animals in some of the evaluated parameters (Table 2). During the last 2 weeks of the experiment, the haematocrit in the infected groups showed lower values than the control group, although without statistical differences. As for total leukocytes, a slight overall increase was observed in challenge control G2 compared to G1 (primary- and challenge-infected), and the same observation was found for monocyte and basophil counts; however, no significant differences could be demonstrated either. Other parameters such as total serum proteins, lymphocyte, eosinophil, and platelet cell counts had approximately the same mean values in the three experimental groups within the sampling period tested.

The body weight of the uninfected controls (G3) increased with respect to both infected groups (G1 and G2) from week 2 after the challenge infection (day 35 of the experiment) until the end of the experimental setting, with significant differences being only recorded at the last body weight measurement (*p* < 0.01) (Figure 3). No significant differences between groups G1 and G2 were found in the intervals assessed.

#### 3.2.3. Histopathological Analysis (Figure 4)

Macroscopic examination at necropsy of the control animals revealed no apparent alteration of small and large intestinal mucosa. However, in both the primary and the challenged *E. christenseni*-infected goat kids, slight hyperaemia and an increase in size of both the small and large intestine were observed.

**Figure 4 animals-15-00139-f004:**
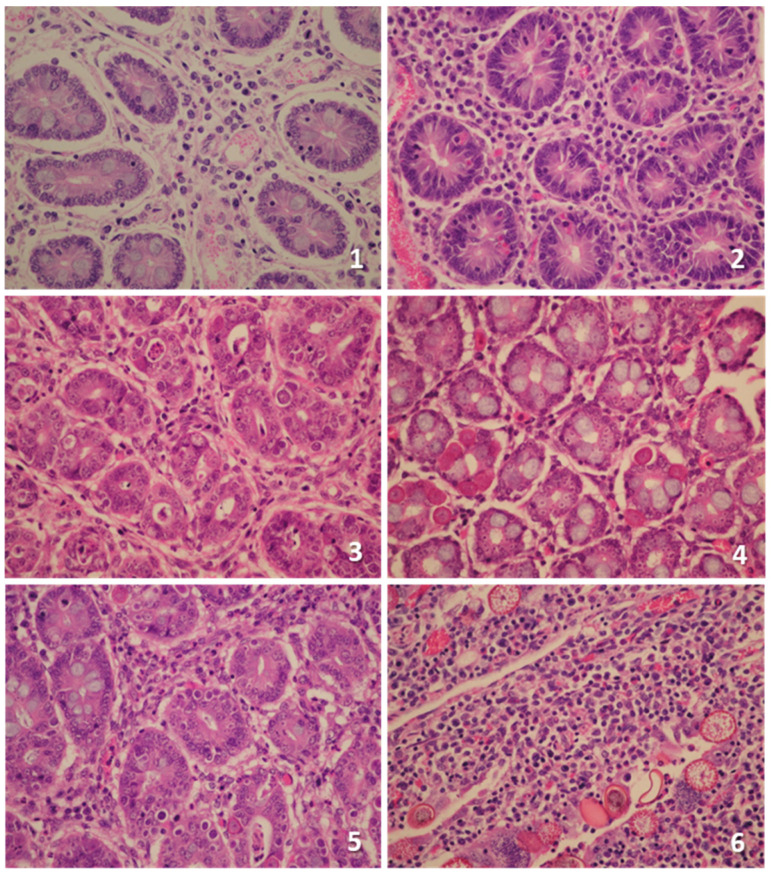
Main histopathological findings in the intestinal mucosa of goat kids experimentally infected with *E. christenseni*. (**1**) Prominent hyperaemia and interstitial oedema of the lamina propria associated with mononuclear infiltration. (**2**) Intense diffuse mononuclear infiltration in the lamina propria. (**3**–**5**) Numerous intracellular macro- and microgametocytes at different stages of development. (**6**) Intense infiltration of inflammatory cells, rich in leucocytes eosinophils, associated with intracellular gamonts, schizonts, and oocysts in the lumen of necrotic glands.

As shown in Table 3, the score for microabscesses was significantly higher in the infected groups when compared to controls in all assessed intestinal sections (*p* < 0.05–00.1). Values recorded in the reinfected group G1 were particularly high in the small bowel, but no significant differences were found with respect to challenge controls (G2). On the other hand, goat kids primary-infected at 6 weeks of life (G2) showed a higher intensity of infection than G1 in all intestinal sections, being especially evident in the ileum, while no intracellular parasitic stages (i.e., sporozoites, trophozoites, (macro)schizonts, gamonts) were recorded in control group G3.

In general, the necrosis degree of the intestinal mucosa was very low in all examined groups, being practically absent at the level of the ileum and colon. In the jejunum, slightly higher values were observed in group G2, although detected differences were not significant. On the contrary, in all the intestinal sections analysed, the reinfected group G1 showed hyperaemia values higher than the rest of the groups, the differences observed being particularly significant when compared with control animals, which did not show evidence of hyperaemia.

When comparing the size of ileal lymphoid follicles, slightly larger values were observed in the infected groups when compared with the control one, particularly in primary-infected group G2. A similar pattern was noted for the size/thickness of the intestinal mucosa but only in ileal and colonic samples. As for other parameters, significant differences between groups could not be demonstrated either. Along the same lines, inflammatory cell counts (cellular infiltrates) were increased in the infected groups G1 and G2 in all the intestinal sections examined when compared to uninfected controls, with significant differences (*p* < 0.05–0.001). No statistical discrepancies were found between both infected groups. Finally, the size of the mesenteric lymph nodes of G1 and G2 showed higher values than that of the control group but without significant differences.

To have an overall view of the histopathological data, the summation of the inflammatory parameters and the lesions of each group were evaluated for the three intestinal sections examined. For this, all parameters were scored as explained previously. When comparing the summation of all the parameters analysed, the scores were higher in the infected groups G1 and G2 with respect to controls in the three intestinal sections. Additionally, challenge-infected G1 showed an increased score compared to challenge control group G2 but only in ileal and jejunal samples (Table 3). Approximately the same was observed for immunological parameters (i.e., microabscesses, hyperaemia, ileal lymphoid follicular sizes, and inflammatory cell counts), but in this case, values for the two infected groups were rather similar (Table 3). No additional information was found when considering jejunal, ileal, and colon sections altogether.

## 4. Discussion

In the present work, we successfully isolated a novel *Eimeria christenseni* strain (GC) from faecal samples of naturally infected goat kids from Gran Canaria (Spain). This field strain presents similar morphological and biological characteristics to those previously described by other authors [46]. In addition, we proved that the *E. christenseni* GC strain is able to infect goat kids at early stages but without apparent clinical signs and to induce protective immune responses in experimental infections.

The caprine species *E. christenseni* was first described in 1962 [46]. Previously, oocysts of this *Eimeria* species were considered to belong to *Eimeria ahsata*, a pathogenic species that affects sheep, but it was later demonstrated that the morphological characteristics of both species differ in some respects and that monoxenous *E. christenseni* infects exclusively goats. Moreover, recent studies on *E. christenseni* morphology have reported an average oocyst size of 34.5 × 23.3 µm (28.9–35.8 × 16.4–25.8 µm), with a length-to-width ratio of 1.5 and an ovoid or ellipsoid shape [47]. A further evaluation of morphological characteristics of *E. christenseni* showed that the oocyst size ranged between 38.4 ± 2.7 µm and 24.7 ± 2.2 µm for the longitudinal and transversal diameter, respectively, while mean size of sporocysts was estimated at 13.9 ± 1.5 µm × 9.1 ± 0.8 µm [48]. In the current study, oocysts were slightly larger for a total of 200 oocyst measurements, a greater sample size than the 35 and 49 oocysts analysed by Al-Habsi et al. [47] and Macedo et al. [48], respectively. Morphologically, *E. christenseni* oocysts are larger and more ovoid-shaped (longer than wide), commonly with a pear shape, compared with other *Eimeria* species of goats. In addition, their sporocysts are larger than the rest of the caprine *Eimeria* spp. [48]. The micropylar cap, which is quite prominent according to previous descriptions [47,48], may occasionally become detached, as mentioned in Section 3.

The prepatent period of *E. christenseni* was slightly shorter in animals infected at 2 weeks of age than in those infected at 6 weeks of age, which agrees with previous findings showing that younger goat kids experimentally infected with *E. ninakohlyakimovae* have a longer prepatent period [27]. In general, these results are in accordance with those reported by Lima [19] in 2–4-week-old goats, where the prepatent period of *E. christenseni* was estimated at 17 (14–23) days p. i. Prepatency of *E. christenseni* therefore resembles that of other pathogenic *Eimeria* spp. such as *E. ninakohlyakimovae* where oocyst excretion begins at 14–15 days p. i. [37] or *E. arloingi* with a prepatent period of 16–18 days [49]. Hence, it would be reasonable that in mixed *Eimeria* spp. infections, goats start releasing oocysts of these three pathogenic species two weeks after oral infection, as described previously [50,51]. In the present study, oocyst shedding progressively increased to peak values on days 24–25 p. i. and then progressively decreased. These results are somewhat different from another study, where *E. christenseni* oocyst shedding remained high until day 28 p. i. without recording peak OPG levels [19]. The excretion dynamics of *E. christenseni* differs from that described for other species such as *E. ninakohlyakimovae*, for which the OPG peak is reached at 17–18 days p. i., that is, 3–4 days after the beginning of the patent period [27].

In natural field infections, it is common for an animal to be simultaneously infected by several *Eimeria* species [2]. The dynamic of oocyst excretion in those conditions differs from what would be expected according to the prepatent periods described for the different *Eimeria* species referred to above. Thus, in a previous study trying to evaluate toltrazuril efficacy in goat kids affected by coccidiosis, the frequency of *E. christenseni* was close to 100% and significantly higher than those recorded for *E. ninakohlyakimovae* and *E. arloingi,* at least until goat kids were 5 weeks old [52], even though the latter two *Eimeria* species would have an even shorter prepatent period than *E. christenseni*. Early oocyst appearance of *E. christenseni* has been also encountered in other field studies on coccidiosis in goat kids [18,53]. These observations were the reason why in the present study the animals were infected at 2 weeks of age, earlier than in experimental infections previously performed by our research group with *E. ninakohlyakimovae* [27,37,54].

*E. christenseni* is considered among the three most pathogenic *Eimeria* species in goats, after *E. ninakohlyakimovae* and *E. arloingi*. Two-to-four-week-old goat kids infected with 1–5 × 10^5^ sporulated oocysts were found to produce severe diarrhoea, anorexia, polydipsia, and extreme weakness [19]. By contrast, in the present study, no apparent clinical signs were demonstrated in animals infected with the *E. christenseni* GC strain. As animals of approximately the same age were employed in both studies, discrepancies would be related to other factors, such as the pathogenicity of the strain itself. Considering that the same dose employed here (2 × 10^5^) for experimental infection with *E. ninakohlyakimovae* [29,37], or even lower for *E. arloingi* (1 × 10^3^–1 × 10^5^) [49], may cause clinical signs from mild to severe, it is likely that the severe *E. christenseni*-induced coccidiosis described by Lima [19] could be related to the higher pathogenicity of the strain used in their study, although the involvement of other, still unknown factors should not necessarily be ruled out as reported for other ruminant species. The *E. christenseni* GC strain isolated here would then be considered as a caprine *Eimeria* strain of low pathogenicity, as despite not producing clinical signs, it showed a quite long-term period of oocyst excretion, as well as a very high oocyst production, reaching several millions per gram of faeces in some animals.

Coinciding with the absence of clinical signs, no relevant changes were found in haematological parameters, which remained within the reference values for goats. However, slight non-significant differences could be observed between the infected and control groups, particularly at the end of this study. Similarly, in *E. ninakohlyakimovae* experimental infections, despite the severity of clinical signs, haematological alterations were not as striking as expected either [37,55]. In these two studies, the most relevant alteration was an increase in the haematocrit parameter, probably due to a decrease in the total volume of circulating blood because of blood losses during haemorrhagic diarrhoea, not observed in the *E. christenseni* experimental goat kids’ infections conducted here.

At necropsy, the intestines of *E. christenseni*-infected animals showed no significant macroscopic alterations, in agreement with that reported in goat kids experimentally infected with *E. ninakohlyakimovae* [37]. Probably, since the animals were euthanised in both cases in the postpatent period (i.e., 21 and 31 days after the first oocyst excretion of *E. ninakohlyakimovae* and *E. christenseni,* respectively), most of the gross lesions would have disappeared. However, histology still revealed the presence of moderate alterations in the small and large intestine compared to uninfected controls, as well as scarce parasitic load, most of them corresponding to sexual macro- and macrogamont stages. Thus, as previously reported in experimental infections with *E. ninakohlyakimovae* [37,56], there was a hyperplasia of the intestinal epithelium and hypertrophy of the ileal lymphoid follicles (Peyer’s patches) and mesenteric lymph nodes. Furthermore, confirming the existence of an inflammatory response, a significant cellular infiltrate was found in all the intestinal sections examined, together with hyperaemia, microabscesses, and, to a lesser extent, mucosal necrosis. Confirming what was discussed above, the total histology score was significantly higher in both *E. christenseni*-infected groups compared to uninfected controls, with slight differences being recorded between primary- and challenge-infected animals.

Previous studies by Lima [19] showed that goats experimentally infected with *E. christenseni* grew more slowly than uninfected control animals. This observation coincides with the results obtained here, where the infected animals, independently of the infection age, showed a body weight (BW) below that of the uninfected goat kids, especially noticeable at the end of the experiment. However, the difference in BW between the infected and uninfected animals was not as pronounced as that described for other *Eimeria* species, such as *E. ninakohlyakimovae*, where the growth rate of the infected animals may decrease up to 15% and more than 10% during the whole patent phase [37]. Usually, delayed growth rates in goat coccidiosis are correlated with the presence of clinical signs [52]. As diarrhoea or other common signs of clinical manifested coccidiosis were not found here, the reduced BW of *E. christenseni*-infected animals compared to controls would be due to subclinical coccidiosis, which could be assumed due to the existence of parasitic forms in gut mucosa throughout this study and the related morphological changes of the intestinal mucosa.

Goat kids first infected at 2 weeks of age and challenged had a significant reduction in the number of OPG when compared to challenge controls, where the number of OPG was higher at all sampling times. This reduction in oocyst excretion provides evidence that the *E. christenseni* GC strain can indeed induce a protective immune response, a phenomenon well documented in other *Eimeria* spp. affecting ruminants [37,57,58]. Accordingly, the total histology score for immunity-related parameters was slightly higher in challenge-infected than in challenge-control animals, a finding commonly found in experimental infections with *E. ninakolhyakimovae* [27,29,37]. By contrast, when considering total peripheral leukocyte counts (in particular monocytes and basophils), a slight overall increase was observed in challenge controls compared to reinfected animals, suggesting that before the development of a protective local immune response, a transient recruitment of innate effector immunocompetent cells such as neutrophils and eosinophils is required during ruminant eimeriosis in vivo [27,59].

## 5. Conclusions

A new field strain of *Eimeria christenseni* (named *E. christenseni* GC strain) was isolated in the present study, which demonstrated morphological and life cycle characteristics like those previously reported for this species. Despite intense oocyst excretion, no apparent clinical signs were found, so the pathogenicity of this parasite strain can be considered very low, in contrast with other species in goats, such as *E. ninakohlyakimovae* and *E. arloingi*, which also develop the first schizogony in the ileal lacteal endothelium. The *E. christenseni* GC strain would then be of great interest for comparative studies focused on the development of new control strategies against caprine coccidiosis and the better understanding of parasite–host cell interactions.

## Figures and Tables

**Figure 1 animals-15-00139-f001:**
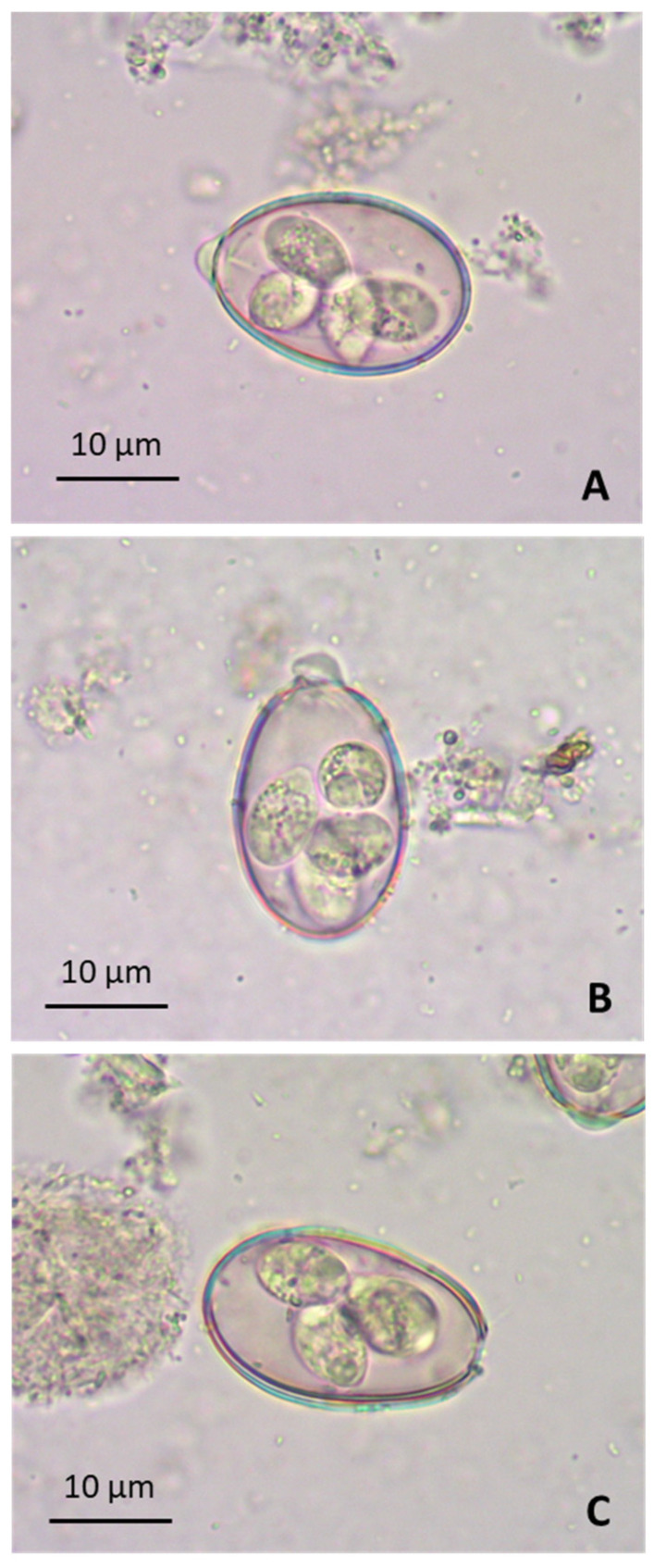
Morphological characteristics of oocysts from the *E. christenseni* GC strain. (**A**) Full sporulated oocyst. (**B**) Sporulated oocyst with semidetached micropillar cap. (**C**) Sporulated oocyst with completely detached micropillar cap.

**Figure 3 animals-15-00139-f003:**
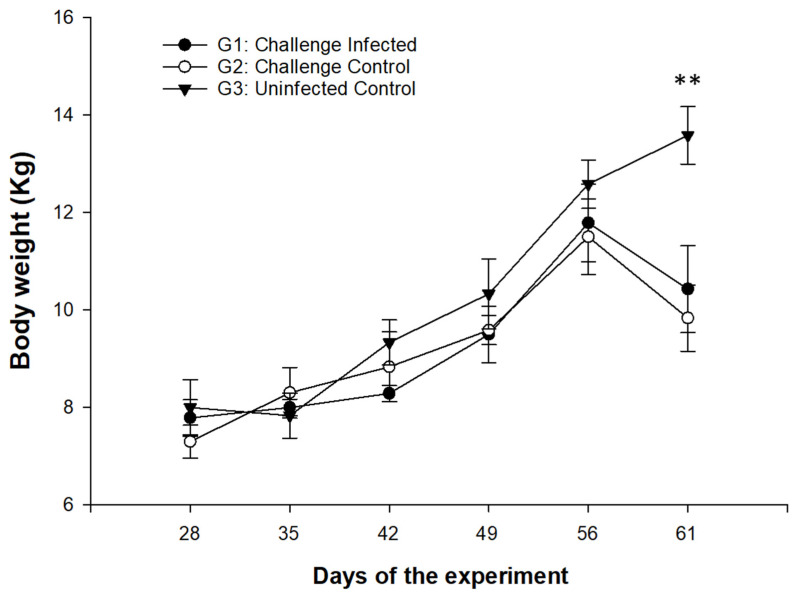
Weekly variation in body weight in goat kids challenge-infected (G1) and primary-infected (G2) with *E. christenseni* GC strain compared to uninfected controls (G3). Data are expressed as mean ± SEM. (**) *p* < 0.01.

**Table 1 animals-15-00139-t001:** Morphological characteristics of the GC strain of *Eimeria christenseni*.

	Oocyst Longitudinal Diameter (µm)	Oocyst Transversal Diameter (µm)	Micropillar Cap Thickness (µm)	Sporocyst Longitudinal Diameter (µm)	Sporocyst Transversal Diameter (µm)
**Average**	40.46	27.30	2.06	16.03	9.61
**Standard deviation**	1.93	1.63	0.41	0.80	0.48
**Standard** **error**	0.14	0.12	0.03	0.056	0.03
**Maximum**	46.00	32.05	3.54	19.1	11.62
**Minimum**	36.17	23.17	1.13	13.51	8.61

**Table 2 animals-15-00139-t002:** Haematological parameters in *Eimeria christenseni* primary- and challenged-infected goat kids.

	HTO (SEM)%	TPR (SEM) g/dL	WBC (SEM) cells/µL	NEU (SEM) cells/µL	LYM (SEM) cells/µL	EOS (SEM) cells/µL	MON (SEM) cells/µL	BAS (SEM) cells/µL	PLA (SEM) cells/µL
**Days of the experiment**	**Group 1: primary infection at 2 weeks of age and challenge infection at 6 weeks of age**								
28 *	33.5 (1.5)	5 (0.1)	6.7 (1.1)	2.2 (0.6)	4.4 (0.9)	0.2 (0.1)	0.01 (0)	0.09 (0.09)	431.8 (56.9)
35	36.5 (1.1)	5.4 (0.1)	6.4 (0.6)	2.6 (0.5)	3.5 (0.7)	0.3 (0.1)	0.01 (0)	0 (0)	561.5 (42.3)
42	30.6 (1.2)	5.2 (0.2)	7 (0.8)	2.4 (0.4)	4.3 (0.4)	0.2 (0.1)	0.02 (0)	0.11 (0.08)	465.3 (78.2)
49	29.1 (1.9)	4.7 (0.4)	8.4 (1.1)	2.5 (0.6)	5.6 (0.8)	0.2 (0)	0.05 (0.01)	0.05 (0.02)	657.9 (65.1)
56	27.6 (1.8)	4.8 (0.2)	10.8 (1.2)	3.6 (0.8)	6.5 (0.9)	0.2 (0.1)	0.24 (0.13)	0.19 (0.1)	620.3 (58.3)
61	25.3 (1.6)	5 (0.3)	12.6 (1.1)	4.8 (0.6)	7.2 (0.7)	0.1 (0)	0.4 (0.16)	0.07 (0.04)	536.3 (132.8)
**Days of the experiment**	**Group 2: primary infection at 6 weeks of age**								
28 *	33.8 (1.6)	5.3 (0.2)	8.2 (1.2)	3.1 (0.7)	4.6 (1.2)	0.3 (0.1)	0.02 (0)	0.31 (0.13)	484.8 (42.1)
35	38 (1.9)	5.7 (0.1)	9.3 (1.6)	3.2 (0.7)	5.4 (1.6)	0.4 (0.1)	0.02 (0)	0.23 (0.19)	448.5 (97.4)
42	36.3 (0.9)	5.5 (0.1)	9.9 (1.6)	3.7 (0.6)	5.6 (1.4)	0.3 (0.1)	0.06 (0.04)	0.23 (0.22)	482.8 (71.8)
49	33.1 (3.8)	4.7 (0.3)	9.4 (1.2)	2.8 (0.5)	5.9 (0.9)	0.2 (0)	0.19 (0.19)	0.2 (0.19)	672 (109.1)
56	28.8 (7.4)	4.5 (0.7)	13.1 (4.6)	5.9 (2.8)	6.2 (1.7)	0.1 (0)	1.61 (1.1)	0.04 (0.02)	550.25 (160.4)
61	27.7 (9.1)	5 (0.4)	11.7 (2.5)	2.5 (1.1)	7.5 (0.8)	0.1 (0.1)	0.83 (0.6)	0.23 (0.2)	618.7 (138.6)
**Days of the experiment**	**Group 3: uninfected controls**								
28 *	32.2 (2.2)	5 (0.2)	6.3 (0.9)	1.7 (0.2)	4.3 (0.9)	0.3 (0.1)	0.01 (0)	0 (0)	406 (26.6)
35	35.5 (3.2)	5.3 (0.2)	6.8 (1.6)	2 (0.5)	4.3 (1.1)	0.3 (0.1)	0.02 (0.01)	0.12 (0.12)	364.2 (78.5)
42	33.3 (1.1)	5.1 (0.1)	8.6 (1.2)	2.8 (0.5)	5.3 (0.8)	0.4 (0.1)	0.01 (0)	0.16 (0.09)	371.7 (23.2)
49	32.2 (0.8)	5.3 (0)	8.9 (1.1)	3.2 (0.7)	5.2 (0.6)	0.4 (0.1)	0.02 (0)	0.02 (0)	399.8 (36)
56	34.7 (2.1)	5.3 (0.1)	10.1 (1.8)	4.2 (1.2)	5.5 (0.8)	0.1 (0)	0.23 (0.16)	0.08 (0.03)	433.5 (25.6)
61	31.3 (2.4)	4.5 (0.1)	9.6 (1.3)	3.6 (1)	5 (0.6)	0.1 (0)	0.76 (0.3)	0.11 (0.06)	494.8 (39.4)

* Day of challenge infections; TPR: total proteins; HTO: haematocrit; WBC: while blood cells; NEU: neutrophils; LYM: lymphocytes; MON: monocytes; EOS: eosinophils; BAS: basophils; PLA: platelets. The results are expressed as the mean ± SEM (standard error of the mean) of different days of the experiment.

**Table 3 animals-15-00139-t003:** Histopathological parameters in *Eimeria christenseni* primary- and challenge-infected goat kids.

	Jejunum—JE			Ileum—IL			Colon—CO		
	G1	G2	G3	G1	G2	G3	G1	G2	G3
**Microabscesses**	1.7 (0.5)	1.3 (0.2)	0.0 (0.0)	1.3 (0.20)	1.0 (0.0)	0.3 (0.4)	0.9 (0.2)	1.0 (0.0)	0.0 (0.0)
**Infection intensity**	2.0 (0.3)	2.5 (0.4)	0.0 (0.0)	0.6 (0.3)	1.5 (0.3)	0.0 (0.0)	0.9 (0.3)	1.3 (0.4)	0.0 (0.0)
**Necrosis**	0.3 (0.2)	0.5 (0.2)	0.0 (0.0)	0.1 (0.2)	0.0 (0.0)	0.0 (0.0)	0.0 (0.0)	0.2 (0.2)	0.0 (0.0)
**Mucosal hyperaemia**	1.7 (0.2)	1.3 (0.2)	0.0 (0.0)	1.7 (0.2)	1.2 (0.2)	0.0 (0.0)	1.7 (0.2)	1.0 (0.0)	0.0 (0.0)
**Ileal follicular size**				3.6 (0.5)	4.4 (0.5)	3.1 (0.5)			
**Mucosal thickness**	732.7 (71.4)	904.0 (69.5)	849.3 (52.8)	680.0 (77.5)	546.0 (71.6)	448.0 (66.0)	606.7 (58.7)	457.3 (83.2)	382.7 (13.20)
**Inflammatory cell count**	140.1 (14.7)	113.9 (9.4)	82.5 (12.8)	115.5 (9.0)	104.3 (10.1)	52.5 (5.8)	97.6 (7.9)	118.1 (15.7)	36.0 (4.8)
	**G1**			**G2**			**G3**		
**Follicular size of mesenteric lymph nodes**	3.9 (0.3)			3.7 (0.1)			2.7 (0.2)		
	**JE**	**IL**	**CO**	**JE**	**IL**	**CO**	**JE**	**IL**	**CO**
**All parameters ^[a]^**	8.0	8.6	4.9	7.3	7.9	5.2	2.3	3.7	1.0
**Immunological-related parameters ^[b]^**	5.8	5.9	3.9	4.3	5.4	3.7	2.3	3.7	1.0
**All intestinal sections and parameters ^[c]^**	22.7			21.6			6.3		
**All intestinal sections and immunological parameters ^[d]^**	16.9			14.6			6.3		

**G1**: goat kids primary-infected at 2 weeks of age and challenged-infected at 6 weeks. **G2**: primary-infected at 6 weeks of age (challenge control). **G3**: non-infected animals. *Microabscess score*: (0) absence; (1) few, showing crypts with only one or two inflammatory cells inside; (2) moderate, showing all crypts with numerous cells inside; (3) severe. *Infection intensity*: (0) absence of parasites; (1) few and isolated parasites; (2) infection foci are evident; (3) the infection is widespread, affecting the mucosa and submucosa. *Necrosis score*: (0) absence; (1) necrosis in half of the microvilli; (2) necrosis in all the microvilli. *Hyperaemia score*: (0) absence; (1) moderate; (2) severe. *Ileal follicular size*: µm. *Mucosal thickness*: µm. *Inflammatory cell count*: cell/mm^2^. ^[a]^ Sum of the total score for microabscesses, infection intensity, necrosis, hyperaemia, ileal follicular size, mucosal thickness, inflammatory cell counts. ^[b]^ Sum of the total score for microabscesses, hyperaemia, ileal follicular size, inflammatory cell counts. ^[c]^ Sum of the total score of the three intestinal sections (JE: jejunum, IL: ileum, CO: colon) + score for the follicular size of mesenteric lymph nodes. ^[d]^ Immunological parameters considering all intestinal sections. Data are expressed as mean values (± SEM, Standard Error of the Mean) or average scores.

## Data Availability

All data presented here are available upon request.

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
