# Peer review of "Isolation of a Novel Caprine Eimeria christenseni Strain (GC) in Canary Islands and Analysis of Parasitological, Clinical, and Pathological Findings on Experimentally Infected Goat Kids"

_animals, 2025, doi:10.3390/ani15020139_

Round 1

Reviewer 1 Report

Comments and Suggestions for Authors

The manuscript 'Isolation of a novel caprine Eimeria christenseni strain (GC) in the Canary Islands and analysis of parasitological, clinical and pathological findings on experimentally infected goat kids' deals with a very important animal health issue for goat breeding worldwide and the resulting data may be useful in the implementation of coccidiosis control in goats.

The manuscript is well written, all the different sections are adequately described, the results are accompanied by graphs and tables and the data are analysed using appropriate statistical methods. The bibliography provided is also relevant to the topic, although some of the bibliographical entries (line 85) on the control of coccidiosis by phytotherapy could be improved by adding some more recent entries on the control of coccidia in kids.

With regard to materials and methods, and in particular paragraph 2.6, for the part on histopathological analyses, I would advise the authors to include the descriptive part of the various parameters used, and in particular the scores, by removing them from the heading of Table 3. This would make the table easier to understand.

Author Response

REVIEWER 1 

1) The manuscript 'Isolation of a novel caprine Eimeria christenseni strain (GC) in the Canary Islands and analysis of parasitological, clinical and pathological findings on experimentally infected goat kids' deals with a very important animal health issue for goat breeding worldwide and the resulting data may be useful in the implementation of coccidiosis control in goats. 

We thank the reviewer for all the time employed and valuable consideration, comments and suggestions. 

2) The manuscript is well written, all the different sections are adequately described, the results are accompanied by graphs and tables and the data are analysed using appropriate statistical methods. The bibliography provided is also relevant to the topic, although some of the bibliographical entries (line 85) on the control of coccidiosis by phytotherapy could be improved by adding some more recent entries on the control of coccidia in kids. 

The two references on phytotherapy referred to lamb coccidiosis have been replaced by more recent ones on goat coccidiosis: 

  1. Daiba, A.R.; Kagira, J.M.; Ngotho, M.; Kimotho, J.; Maina, N. In vitro anticoccidial activity of nanoencapsulated bromelain against Eimeria spp. oocysts isolated from goats in Kenya. Vet. World. 2022, 15, 397–402. doi: 10.14202/vetworld.2022.397-402. 
  2. Neha, A.; Shaik, A.; Chelkapally, S.C.; Kolikapongu, R.S.; Namani, S.C.; Erukulla, T.; Batchu, P.; Mendez, N.; Smith, Y.; Brown, D.; Whitley, N.C.; Pech-Cervantes, A.A.; Dykes, G.S.; Owen, V.R.; Kannan, G.; Miller, J.E.; Siddique, A.; Terrill, T.H. Effect of feeding a blackseed meal-sericea lespedeza leaf meal pellet on gastrointestinal nematode and coccidia infection and animal performance in young goats. Vet. Parasitol. 2024, 331:110253. doi: 10.1016/j.vetpar.2024.110253.

3) With regard to materials and methods, and in particular paragraph 2.6, for the part on histopathological analyses, I would advise the authors to include the descriptive part of the various parameters used, and in particular the scores, by removing them from the heading of Table 3. This would make the table easier to understand. 

Lines 232-233: following the recommendation of Reviewer 3, the information about the scores has been removed from the M&M and a reference to it has been made “as described in Table 3”. We think that for readers it would easy to understand Table 3 if score date are in the footnotes of the table. 

Reviewer 2 Report

Comments and Suggestions for Authors

Dear authors,

I would like to commend you on the high quality of your handwriting. The article is well written, clear and provides a thorough analysis of the topic.

The present study describes a new strain of coccidia of the genus Eimeria.  It addresses not only the detailed morphological description but also the effect on animal health. 

The topic is relevant to the field. The study contains a thoroughly developed methodology with relevant results that are well described and explained. Coccidia of the genus Eimeria in small ruminants are not often studied or morphologically described. The material provided in this study is therefore needed. 

The introduction is well structured and provides a solid background and rationale for the study. The methodology is described in sufficient detail to ensure reproducibility and transparency, and the results are presented comprehensively with relevant statistical analyses.

In addition, the discussion successfully places the findings in the context of the existing literature and highlights their significance and potential implications. The conclusions are well supported by the data and the manuscript as a whole represents a valuable contribution to the field.

The references used are relevant and fully meet the focus of the submitted publication. 

I have no major comments on the content or presentation of the manuscript. My minor comments, if any, are listed below for consideration, although they in no way detract from the overall quality of the work.

1. Could you please provide more specific information on how the faecal samples were collected from the naturally infected animals? It would be helpful to understand whether the sampling was conducted directly from the animals, from the environment, etc.

2. Could you clarify the analytical sensitivity of the McMaster method used in your study?

Author Response

REVIEWER 2 

I would like to commend you on the high quality of your handwriting. The article is well written, clear and provides a thorough analysis of the topic. 

The present study describes a new strain of coccidia of the genus Eimeria.  It addresses not only the detailed morphological description but also the effect on animal health.  

The topic is relevant to the field. The study contains a thoroughly developed methodology with relevant results that are well described and explained. Coccidia of the genus Eimeria in small ruminants are not often studied or morphologically described. The material provided in this study is therefore needed.  

The introduction is well structured and provides a solid background and rationale for the study. The methodology is described in sufficient detail to ensure reproducibility and transparency, and the results are presented comprehensively with relevant statistical analyses. 

In addition, the discussion successfully places the findings in the context of the existing literature and highlights their significance and potential implications. The conclusions are well supported by the data and the manuscript as a whole represents a valuable contribution to the field. 

The references used are relevant and fully meet the focus of the submitted publication.  

I have no major comments on the content or presentation of the manuscript. My minor comments, if any, are listed below for consideration, although they in no way detract from the overall quality of the work. 

We thank the reviewer for all the time employed and valuable consideration, comments and suggestions. 

1) Could you please provide more specific information on how the faecal samples were collected from the naturally infected animals? It would be helpful to understand whether the sampling was conducted directly from the animals, from the environment, etc. 

The way of sampling for naturally infected animals has been specified: 

Lines 132-133: “individually” has been replaced by “by rectal route” 

2) Could you clarify the analytical sensitivity of the McMaster method used in your study? 

The sensitivity of the McMaster method has been included: 

Lines 155-156: “with a sensitivity of ≥ 100 OPG” 

Reviewer 3 Report

Comments and Suggestions for Authors

The manuscript “Isolation of a novel caprine Eimeria christenseni strain (GC) in Canary Islands and analysis of parasitological, clinical and pathological findings on experimentally infected goat kids” provides information about the characterization of a new strain of E. christenseni. I consider that the data included is of interest to the scientific community and it may be useful for developing new preventive methods such as vaccines. Nevertheless, there are some shortcomings that must be corrected before being accepted for publication:

Major concerns:

1.-The statistical analysis section is poor. Some considerations:

-Line 244: “Faecal oocyst counts were logarithmically transformed and added by one”. I understand that you first calculated the logarithm and then added 1. Maybe it is easier to state only: “Faecal oocyst counts were logarithmically transformed (log[OPG+1])…”.

-Line 245: Kolmogorov-Smirnov normality test was employed for checking if variables are normally distributed.

-Line 247: The formula is not easy to interpret. Maybe something is wrong or is not present.

-Lines 248-250: Please, state what you used each of the tests for.

2. Move Tables and Figures close to the text referring to them.

3. Considering the results (especially those in Figure 2), maybe would be interesting to have another group: animals infected with oocysts at 2 weeks of age, without challenge infection at 6 weeks. Could the challenge infection have some side effect on the primary infection? Maybe stress after inoculation (or other reasons) might affect to the opg results of the primary infection…

Other comments:

-Line 172: Can you get reliable measures of sporozoites and micropyle cap using 100x and 400x magnification? Do not use “10x and 40x objectives”, use “100x and 400x magnification”. We are assuming that you are using 10x ocular lens. But maybe not.

-Line 181: “their purity was close to 100%” and line 255 “average time was about 7 days”. Be exact.

-Lines 223-241: It would be better if this information was on one Table. It is also in the footnote of Table 3, so maybe you can refer to it.

-Lines 273-282. This is difficult to read and understand, because you are referring to the same period twice (lines 273-275 and 275-279).

-Figure 2. Delete (OPG) in text and Figure, since it is always together to “oocysts per gram of faeces”. What is SEM? Define it. Is the asterisk related to significant differences? No significant differences were found then?

-Table 2. Although the number in parentheses below the main number is defined in the footnote, I would include it in the header :  HTO% (SE).

-Table 3: State that the results are averages of scores

-Line 454-455: The micropylar cap is not the same as the micropyle…

-Line 456: May this lead to a misidentification of species?

Line 487: Delete “sporulated oocysts”. It is repeated.

Author Response

REVIEWER 3 

The manuscript “Isolation of a novel caprine Eimeria christenseni strain (GC) in Canary Islands and analysis of parasitological, clinical and pathological findings on experimentally infected goat kids” provides information about the characterization of a new strain of E. christenseni. I consider that the data included is of interest to the scientific community and it may be useful for developing new preventive methods such as vaccines. Nevertheless, there are some shortcomings that must be corrected before being accepted for publication. 

We thank the reviewer for all the time employed and valuable consideration, comments and suggestions. 

Major concerns: 

1) The statistical analysis section is poor. Some considerations: 

-Line 244: “Faecal oocyst counts were logarithmically transformed and added by one”. I understand that you first calculated the logarithm and then added 1. Maybe it is easier to state only: “Faecal oocyst counts were logarithmically transformed (log[OPG+1])…”. 

“and added by one” has been deleted from the sentence. 

-Line 245: Kolmogorov-Smirnov normality test was employed for checking if variables are normally distributed. 

Lines 243-245: the sentence has been modified as suggested by the reviewer. 

-Line 247: The formula is not easy to interpret. Maybe something is wrong or is not present. 

Line 246: We apologize, the sign “minus” was missing and it has been included “-” 

-Lines 248-250: Please, state what you used each of the tests for. 

Lines 247-252: According to the suggestion of the reviewer, we have specified what the tests were used for. 

One factorial analysis of variance and Tukey’s multiple comparison test were employed to check statistical differences on OPG counts, body weights and histopathological parameters between groups, while the non-parametric Chi-square was used to analyze the sum of the different scores established for histopathological parameters. The software SigmaPlot 14.5 (California, USA) was employed for all the statistical analyses.  

2) Move Tables and Figures close to the text referring to them. 

Following the Author Intructions of Animals, we employed the “Microsoft Word Template”, where “Figure and Tables” were placed at the end of the Results section. 

We assume that, after the edition of the manuscript in its final version, Tables and Figures are to be placed close to the text referring to them. 

3) Considering the results (especially those in Figure 2), maybe would be interesting to have another group: animals infected with oocysts at 2 weeks of age, without challenge infection at 6 weeks. Could the challenge infection have some side effect on the primary infection? Maybe stress after inoculation (or other reasons) might affect to the opg results of the primary infection… 

The experiment was designed to evaluate the development of a protective immune response and, for this reason, the results were compared between the challenged group and the primo-infected group at 6 weeks or challenge control. 

If the animals infected at 2 weeks had not been re-infected, the oocyst counts would have decreased to practically zero at the end of the experiment. Re-infection obviously modified this trend and possibly kept oocyst counts increased for a longer period of time. As suggested by the reviewer, the extent to which OPG counts were modified could have been determined by including an additional group infected at 2 weeks and not challenged. 

Other comments: 

-Line 172: Can you get reliable measures of sporozoites and micropyle cap using 100x and 400x magnification? Do not use “10x and 40x objectives”, use “100x and 400x magnification”. We are assuming that you are using 10x ocular lens. But maybe not. 

Line 177: “10X and 40X objectives” has been replaced by “100X and 400X magnifications” 

-Line 181: “their purity was close to 100%” and line 255 “average time was about 7 days”. Be exact. 

Line 185: the exact percentage for the purity “99%” has been specified. 

Line 260: the exact sporulation time “7 ± 1 days” has been specified. 

-Lines 223-241: It would be better if this information was on one Table. It is also in the footnote of Table 3, so maybe you can refer to it. 

Lines 232-233: the information about the scores has been removed from the M&M and a reference to it has been made “as described in Table 3”. 

-Lines 273-282. This is difficult to read and understand, because you are referring to the same period twice (lines 273-275 and 275-279). 

Line 274: For clarification, “day 51 of the experiment, just 24 days after challenge infection” has been replaced by “day 51 p. i.” 

Line 275: Challenge infection was performed at day 28 after primary infection; we have corrected it in the text “28”. 

Following the recommendation of the reviewer, to avoid redundancies, the text “showed a progressive decrease during the first 24 days p. i., and then” has been deleted. 

-Figure 2. Delete (OPG) in text and Figure, since it is always together to “oocysts per gram of faeces”. What is SEM? Define it. Is the asterisk related to significant differences? No significant differences were found then? 

To simplify “Oocysts per gram of faeces (OPG) count” has been replaced by “OPG counts” in the figure legend. 

“SEM” has been replaced by “SEM (Standard Error of the Mean)” 

As referred in line 278, significant differences (P<0.05) were found in day 51 of the experiment, so the asterisk (*)should be placed over the bar error of this day. My apologies, I am not able in this moment to place the asterisk in the right place. 

As well, (**) (P<0.01) should have been placed over day 61 in Figure 3. 

-Table 2. Although the number in parentheses below the main number is defined in the footnote, I would include it in the header :  HTO% (SE). 

“SEM” (Standar Error of the Mean) has been included in te header and specified in the footnote. 

-Table 3: State that the results are averages of scores 

This has been stated at the end of the footnote: “Data are expressed as mean values ± SEM) or average scores”. 

-Line 454-455: The micropylar cap is not the same as the micropyle… 

“(syn. micropyle)” has been deleted. 

-Line 456: May this lead to a misidentification of species? 

In the absence of a micropylar cap, differences in shape and size would allow differentiation of E. christenseni from other species. 

Line 487: Delete “sporulated oocysts”. It is repeated. 

“sporulated oocysts” has been deleted.